# Interactive Genetic Algorithm Oriented toward the Novel Design of Traditional Patterns

**Jian Lv, Miaomiao Zhu \*, Weijie Pan and Xiang Liu**

Key Laboratory of Advanced Manufacturing Technology, Ministry of Education, Guizhou University, Guiyang 550025, China; jlv@gzu.edu.cn (J.L.); wjpan@gzu.edu.cn (W.P.); lx518626@163.com (X.L.)

**\*** Correspondence: m15585294390@163.com; Tel.: +86-155-8529-4390

**Abstract:** To create alternative complex patterns, a novel design method is introduced in this study based on the error back propagation (BP) neural network user cognitive surrogate model of an interactive genetic algorithm with individual fuzzy interval fitness (IGA-BPFIF). First, the quantitative rules of aesthetic evaluation and the user's hesitation are used to construct the Gaussian blur tool to form the individual's fuzzy interval fitness. Then, the user's cognitive surrogate model based on the BP neural network is constructed, and a new fitness estimation strategy is presented. By measuring the mean squared error, the surrogate model is well managed during the evolution of the population. According to the users' demands and preferences, the features are extracted for the interactive evolutionary computation. The experiments show that IGA-BPFIF can effectively design innovative patterns matching users' preferences and can contribute to the heritage of traditional national patterns.

**Keywords:** patterns with traditional national characteristics; BP neural network; surrogate model; interactive genetic algorithm; aesthetic evaluation

---

## 1. Introduction

Creating consumers' preferred products to achieve commercial goals is an important aspect of product design [1]. Traditional patterns are an important part of the design of national crafts, and they are mostly designed manually according to the designer's own experience; therefore, the designs of patterns are restricted by the individual's level of experience, which is inefficient. Meeting the various design requirements of traditional crafts is difficult [2]. Batik shape patterns are abundant and diverse, with elegant colors and ethnic-specific features. These patterns are rich in aesthetic, cultural, and re-design values, occupying an important position in the development history of modern textiles around the world. Therefore, the batik shape patterns design methods are important. In the study of pattern designs, shape grammar [3] and the fractal algorithm [4] are usually used to extract and reconstruct patterns. However, both use a combined transformation based on shape rules, and shape rules are finite in number. Batik shape patterns are complicated in composition, difficult to represent using traditional geometry, and there are design styles and cultural connotations in the traditional national designs. It is difficult to retain heritage and meet the market demand in the design of these patterns. To explore the emotions of users and design traditional national patterns that users prefer, in order to fully improve the competitiveness of products, further study is warranted.

Aiming at addressing the deficiency of existing pattern design methods, a novel design method is introduced in this study based on the back propagation (BP) neural network user cognitive surrogate of the interactive genetic algorithm with the individual's fuzzy interval fitness (IGA-BPFIF). The advantages of the method in this paper are as follows: (1) constructing the fitness function based on aesthetic evaluation to form the central value of the fuzzy interval in order to retain and inherit the style characteristics of national patterns and the design of national patterns with greater beauty for users,

(2) considering users' cognitive hesitation to revise the evaluation value to increase the diversity of the evolutionary population, (3) using machine learning to predict the fitness of evolutionary individuals to reduce users' operation fatigue, and (4) the BP neural network is trained according to error back propagation by measuring mean squared error (MSE); the training data and surrogate model are continuously updated to guarantee the precision of the surrogate model. For designers, the method can stimulate design inspiration and improve the speed of design to a certain extent. For enterprises, it plays a crucial role in customization of crafts by meeting customers' needs and preferences.

This article is organized as follows. In Section 2, we present the related work. In Section 3, the proposed method is described. In Section 4, the performed experiments are outlined, along with an analysis of the results obtained. In Section 5, we analyze and discuss the algorithm's performance. Finally, in Section 6, the conclusions and a discussion concerning future work are presented.

## 2. Related Work

Batik is one of the intangible cultural heritages in the world, famous for its long history and cultural connotation. In our previous work, Yuan et al. digitally designed butterfly patterns [5] using a fractal algorithm to enrich the batik patterns, classified batik shape patterns [6], and performed retrieval [7] using a machine learning algorithm. Lv et al. realized batik renderings using an interpolation subdivisions algorithm [8]. In order to meet users' personalized needs for batik shape patterns and expand the batik shape patterns design methods, we propose a novel design method based on the interactive genetic algorithm (IGA) in this paper.

The design methods based on the IGA are one of the most effective solutions for responding to user needs and preferences, achieving remarkable results in modifying and improving designs [9]. IGA is an optimized algorithm that is an extension of the traditional genetic algorithm. Through the integration of human intelligence, the optimization of implicit performance indicators, such as human preferences, cognition, emotions, and psychological characteristics, is effective. Therefore, IGA has been widely used in fields such as education [10], engineering [11], and the arts [12]. However, the uncertainty, vagueness, and gradualism of users' cognitive preferences, with fluctuation and evaluation fatigue, affect the performance of the algorithm to a large extent. There are two aspects to the solutions to this problem. The first is to speed up the convergence of IGA, which reduces the number of individuals evaluated by users, thereby easing user fatigue; and the second is to apply a fitness estimation strategy to predict the remaining evolutionary individuals automatically through a small number of individuals evaluated by users as samples, which reduces user evaluation times to reduce user fatigue.

To increase the convergence speed of IGA, there are two major categories of studies. One involves reducing the noise of evaluation, and prior studies have proposed interest degree [13], trust degree [14], credibility [15], hesitation [16,17], and other cognitive rules to reduce the evaluation noise. The second set of solutions involves improving the genetic operators [18]. For the fitness estimation strategies, there are two main categories of research. Methods, such as elite set [19], directed graph [20], grey model [21], fuzzy range [22], and maximum entropy criterion [23], have been proposed. The other category involves applying the fitness function approximation model. At present, studies have been based on machine learning methods [24,25] and on the collaborative filtering algorithms [26].

Although IGA can respond to users' needs to a certain extent, in traditional interactive genetic algorithms (IGA-T), users' cognition is fuzzy, which leads to the inability to accurately measure evolution individuals' fitness as well as user' aesthetic fatigue. Problems with deviation of evolution direction and lacking evolution with low optimized efficiency easily occur. At present, the IGA mainly focuses on an individual's image evaluation, and studies on the aesthetic evaluation of evolutionary individuals are lacking.

## 3. The Proposed Method

The method in this paper builds a unified architecture that integrates user cognitive characteristics, a machine agent model, and evolutionary knowledge. The fitness is formed with a fuzzy Gaussian function for evolutionary individuals of the quantitative rules of aesthetic evaluation and users' cognitive hesitation. Then the fitness of the fuzzy interval is formed through λ-cut set to depict uncertainty of user cognitive. In general, IGA requires a large number of labeled samples generated by a human user who finds the optimal solution. Therefore, if a learning algorithm that simulates the user cognitive processes will reduce fatigue to a large extent. After learning from the training set, the network can automatically map the relationship between the network's input and output. Therefore, in this paper, when the human user feels fatigue, a stable cognitive surrogate model based on BP neural network is obtained by K-fold cross validation on the basis of the human user's historical evaluation information. After evolution of $M$ generations, the agent model presents $M$ optimal individuals in successive generations and the estimated adaptive values. If the predicted values deviate from the human user's expectations, the user can submit the evaluation again and the system will update the training data set automatically. By measuring MSE, the agent model is updated through the user's preferences. The algorithm process is shown in Figure 1.

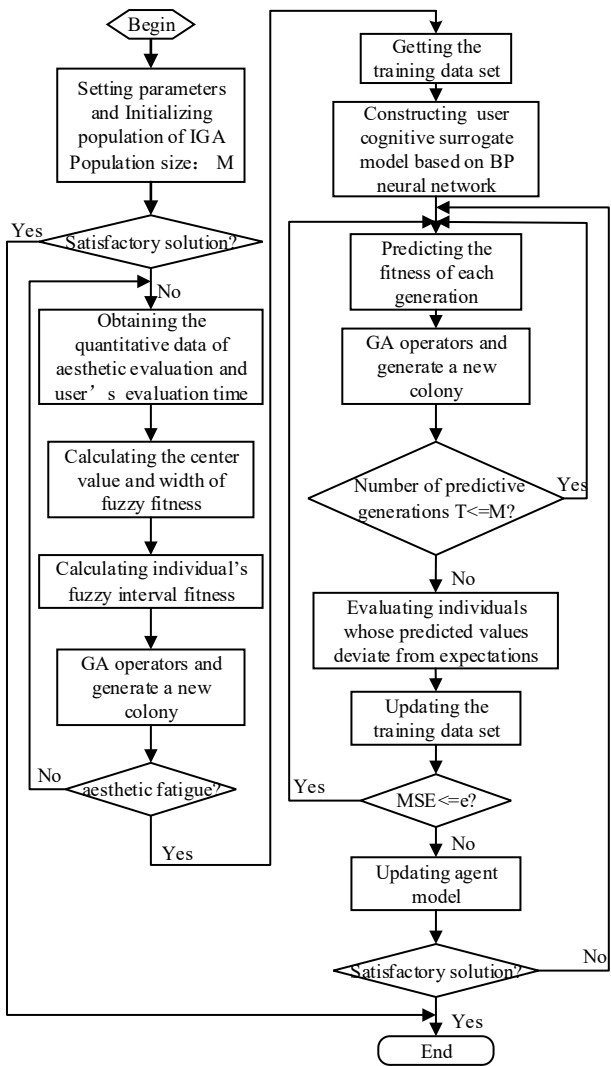

**Figure 1.** The algorithm process.

### 3.1. Individual's Fuzzy Interval Fitness

We constructed a Gaussian membership function with quantifiable rules of aesthetic evaluation as the central value of fuzzy fitness $\widetilde{f}(x_i)$ and the user's hesitation time as the width of $\widetilde{f}(x_i)$, which was transformed into the fuzzy interval fitness through the $\lambda$-cut set. Evolutionary individuals were selected using the interval probabilistic dominant strategy with the league scale of 2 used by Shi et al. [27]. Then, subsequent crossover, mutation, and iteration operations were performed. Patterns can be adjusted to enhance their expression, and finally produce a satisfactory solution.

Without loss of generality, considering an evolutionary individual $\forall x_i \in S$, the domain of the fitness $f(x_i)$ is $[f_a(x_i), f_b(x_i)] \subset R$. The Gaussian membership function of $\widetilde{f}(x_i)$ is shown in Equation (1):

$$\mu_{\widetilde{f}(x_i)}(f(x_i)) = e^{-\frac{1}{2}\left[\frac{f(x_i)-f_c(x_i)}{\sigma(x_i)}\right]} \tag{1}$$

where $f_c(x_i)$ is the central value of $\widetilde{f}(x_i)$, $\sigma(x_i)$ is the width of $\widetilde{f}(x_i)$, and the theoretical domain of the Gaussian membership function is $\mu_{\widetilde{f}(x_i)}(f(x_i)) \subset (0,1]$. Then, the $\lambda$-cut set $f_\lambda(x_i)$ of $\mu_{\widetilde{f}(x_i)}(f(x_i))$ is obtained as the fuzzy interval fitness $\widetilde{F}(x_i)$, as shown in Equation (2):

$$\widetilde{F}(x_i) = \left[f_c(x_i) - \sqrt{-2\ln\lambda}\,\sigma(x_i), f_c(x_i) + \sqrt{-2\ln\lambda}\,\sigma(x_i)\right] \triangleq [F_a(x_i), F_b(x_i)] \tag{2}$$

The central value $f_c(x_i)$ of the evolving individual $x_i$ is composed of several sub-factors $\{f_1, f_2, \ldots f_n\}$ and can be expressed as $f_c(x_i) = \sum\limits_{i=1}^{n} w_i f_i$, where $w_i$ represents the weight corresponding to $f_i$. The characteristics of existing batik patterns were analyzed, and $f_c(x_i)$ was decomposed into three seed factors: style $f_1$, tone $f_2$, and layout $f_3$ as the aesthetic evaluation index. The mean of the field was normalized to $[0, 100]$. $f_1$ is quantified using the user's subjective rating. $f_2$ is quantified using the mean of the color histogram mutual information. Based on the principle of formal aesthetics, $f_3$ is quantified using beauty indicators such as balance, symmetry, and centroid deviation [28], which is $f'_3 = \sum\limits_{i=1}^{3} w_{3i} f_{3i} \subset [0,1]$, where $w_{3i}$ represents the weight corresponding to $f_{3i}$. To maintain the same range of theoretical domain, take $f_3 = 100 f'_3 \subset [0, 100]$.

The degrees of balance represent the difference in the total weight of the patterns on both sides of the horizontal and vertical axes of symmetry; the higher the balance, the calmer the psychological feeling, i.e., $f_{31}$ is closer to 1, and the higher the balance, as shown in Equation (3):

$$f_{31} = 1 - \frac{1}{2}\left(\left|\frac{w_l - w_r}{\max(|w_l|, |w_r|)}\right| + \left|\frac{w_t - w_b}{\max(|w_t|, |w_b|)}\right|\right), w_j = \sum_{i=1}^{n_j} a_{ij} d_{ij}, j = l, r, t, b \tag{3}$$

where $l, r, t, b$ represent the left, right, upper, and lower spaces of the pattern, respectively; $a_{ij}$ represents the space occupied by line $i$ in $j$ space; $d_{ij}$ represents the distance between the centerline of the pattern $i$ and the centerline of the pattern space; and $n_j$ denotes the number of patterns contained in a space. In this paper, the minimum external contiguous rectangle of the contour line is taken as the area of the pattern space.

Symmetry refers to the degrees of symmetry between patterns along the vertical, horizontal, and diagonal lines. The higher the degree of symmetry, the more solemn the psychological feeling, i.e., $f_{32}$ is closer to 1, and the higher the balance, as shown in Equation (4):

$$f_{32} = 1 - \frac{1}{3}(|s_1| + |s_2| + |s_3|) \tag{4}$$

where $s_1, s_2, s_3$ represent the symmetry degree of the vertical, horizontal, and diagonal lines of the pattern, respectively, shown as follows:

$$s_1 = \frac{1}{12}(|x'_{ul} - x'_{ur}| + |x'_{ll} - x'_{lr}| + |y'_{ul} - y'_{ur}| + |y'_{ll} - y'_{lr}| + |h'_{ul} - h'_{ur}| +$$
$$|h'_{ll} - h'_{lr}| + |b'_{ul} - b'_{ur}| + |b'_{ll} - b'_{lr}| + |\theta'_{ul} - \theta'_{ur}| + |\theta'_{ll} - \theta'_{lr}| + |r'_{ul} - \theta'_{ur}| + |r'_{ll} - \theta'_{lr}|)$$

$$s_2 = \frac{1}{12}(|x'_{ul} - x'_{ll}| + |x'_{ur} - x'_{lr}| + |y'_{ul} - y'_{ll}| + |y'_{ur} - y'_{lr}| + |h'_{ul} - h'_{ll}| +$$
$$|h'_{ur} - h'_{lr}| + |b'_{ul} - b'_{ll}| + |b'_{ur} - b'_{lr}| + |\theta'_{ul} - \theta'_{ll}| + |\theta'_{ur} - \theta'_{lr}| + |r'_{ul} - r'_{ll}| + |r'_{ur} - r'_{lr}|)$$

$$s_3 = \frac{1}{12}(|x'_{ul} - x'_{lr}| + |x'_{ur} - x'_{ll}| + |y'_{ul} - y'_{lr}| + |y'_{ur} - y'_{ll}| + |h'_{ul} - h'_{lr}| + |h'_{ur} - h'_{ll}| +$$
$$|b'_{ul} - b'_{lr}| + |b'_{ur} - b'_{ll}| + |\theta'_{ul} - \theta'_{lr}| + |\theta'_{ur} - \theta'_{ll}| + |r'_{ul} - r'_{lr}| + |r'_{ur} - r'_{ll}|)$$

where $x'_j, y'_j, h'_j, b'_j, \theta'_j, r'_j$ represent the dimensionless values after normalization of $x_j, y_j, h_j, b_j, \theta_j, r_j$, respectively, conform to the following constraints:

$$x_j = \sum_i^{n_j} |X_{ij} - X_c|, j = ul, ur, ll, lr$$

$$y_j = \sum_i^{n_j} |Y_{ij} - Y_c|, j = ul, ur, ll, lr$$

$$R_j = \sum_i^{n_j} \sqrt{(X_{ij} - X_c)^2 + (Y_{ij} - Y_c)^2}$$

$$h_j = \sum_i^{n_j} H_{ij}, b_j = \sum_i^{n_j} B_{ij} \theta_j = \sum_i^{n_j} \left| \frac{Y_{ij} - Y_c}{X_{ij} - X_c} \right|$$

$$o'_i = \frac{O_i - \min\limits_{1 \leq j \leq n}\{O_j\}}{\min\limits_{1 \leq j \leq n}\{O_j\} - \min\limits_{1 \leq j \leq n}\{O_j\}}, o = x, y, h, b, \theta, r$$

where *ul*, *ur ll*, and *lr* represent the space on the upper left, upper right, lower left, and lower right of the pattern, respectively; $(X_{ij}, Y_{ij})$ and $(X_c, Y_c)$ represent the coordinates of the pattern *i* in the center of *j* in a certain space and the center of the pattern, respectively; $B_{ij}$ and $H_{ij}$ represent the width and height of the minimum rectangle external to the grain profile, respectively; and $n_j$ denotes the total number of patterns in a space.

Centroidal deviation represents the minimum degrees of the rectangular center between the centroid and contour line. The lower the deviation, the more stable the psychological feeling, i.e., $f_{33}$ is closer to 1, and the higher the balance, as shown in Equation (5):

$$f_{33} = 1 - \frac{1}{2} \left( \left| \frac{\sum\limits_{i=1}^{n} a_i(x_i - x_c)}{\frac{1}{2} \sum\limits_{i=1}^{n} a_i b_w} \right| + \left| \frac{\sum\limits_{i=1}^{n} a_i(y_i - y_c)}{\frac{1}{2} \sum\limits_{i=1}^{n} a_i b_l} \right| \right) \tag{5}$$

where $(x_i, y_i)$ and $(x_c, y_c)$ represent the centroid of the pattern *i* and the center of the smallest rectangle external to the contour of the pattern *i*, respectively; $a_i$ represents the area occupied by pattern; $b_w$ and $b_l$ represent the width and the length of the smallest rectangle external to the contour *i* of the pattern, respectively.

In IGA, users have different degrees of hesitation with different cognition [17]. At the early stage of evolution, the user is not familiar with evolutionary individuals, thus the user has a strong sense of hesitation at this time. Therefore, the fitness of evolutionary individuals has greater uncertainty, so a wider interval value is selected as the fitness of evolutionary individuals. With progressing evolution, user cognition gradually becomes clear and hesitation gradually disappears; therefore, the width of the fitness value interval of an evolutionary individual continually decreases. Therefore, the evaluation

time $T(x_i)$ of individual $x_i$ can be used as the user's cognitive hesitation to describe the width of the fitness value interval. From Equation (2), when $\lambda$ and $f_c(x_i)$ are certain, the width of $\widetilde{F}(x_i)$ is determined by $\sigma(x_i)$. Therefore, $\sigma(x_i)$ is represented by $T(x_i)$ as $\sigma(x_i) = T(x_i)$.

### 3.2. BP Neural Network User Cognitive Surrogate Model

Due to frequent operations intended for human interaction, users easily tire. The construction of the surrogate model is conducive to reducing user fatigue, expanding the search scope of the algorithm, and increasing the diversity of the population. As a non-linear information dynamics model, BP neural networks realize non-linear mapping from input to output and have good fault-tolerant, adaptive, and generalization characteristics. BP neural networks are mainly applied to problems of approximate function, pattern recognition, data compression, and classification. Based on a human user's own historical evaluation information, the surrogate model in this paper obtains the modified fitness using Equation (2) as the labeled samples, adopts K-folded cross validation to train the BP neural network surrogate model, and divides the data into $k$ parts. Each piece of data $D(k) = \{(x_i, \widetilde{F}(x_i)), i = 1, 2, \ldots, N_k\}$ was selected in turn as the test data, and the remaining $k-1$ parts as the training data. As the three-layer BP neural network has a continuous mapping capability from arbitrary input to output, this model adopts the three-layer BP neural network: input layer, hidden layer, and output layer, where $w_{ih}$ and $w_{ho}$ represent the connection weight of neuron nodes in the input layer and hidden layer, hidden layer, and output layer, respectively. The unipolar sigmoid function $f(x) = \frac{1}{1+e^{-x}}$ is used as the transfer function. The number $i$ of the input layer is determined by the component number of the evolving individual $x_i$. The number of nodes of neurons in the output layer $o$ is 2, which represents the upper limit $\bar{F}_a(x_i)$ and lower limit $\bar{F}_b(x_i)$ of the predicted interval fitness $\bar{F}(x_i)$. The number of hidden layer neuron nodes $h$ was determined using $h = \frac{i+o}{2}$ in Lin et al. [29]. $w_{ih}$ and $w_{ho}$ are dynamically adjusted using gradient descent technology, such that the objective function measured by MSE is $MSE = \frac{1}{N_k} \sum\limits_{i=1}^{N_k} \left( \bar{F}(x_i) - \widetilde{F}(x_i) \right)^2$. When the error accuracy requirement is met, the training ends, and the cognitive surrogate model based on BP neural network is obtained. The topology of the surrogate model is shown in Figure 2. The model is used to predict the fitness of subsequent evolution individuals to reduce user operational burden.

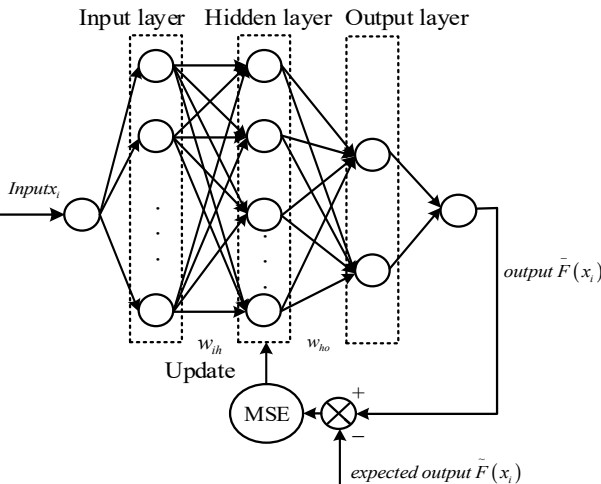

**Figure 2.** Topology of the surrogate model.

As the construction of the surrogate model is based on a human user's own historical evaluation information, with the progress of evolution, user preferences tend to fluctuate, and the fitness of some evolutionary individuals predicted by the surrogate model may deviate from the user expectation. At this point, the surrogate model needs to identify the user preferences to update. As the ultimate goal

of IGA is to search once for the optimal individual surrogate model performances, and the number of evolution generations is $M$ times, where $M$ is the initial population size, the optimal individual of each generation is saved, and $M$ predicted optimal individuals are presented to users. If the predicted fitness of individuals deviates from the user's expectations, the user can re-submit the evaluation data. The evaluation data are included in data set $D(k)$ as the sample data randomly, and the training sample data are verified through K-fold cross validation. If $MSE \geq e$, where $e$ is error accuracy, the surrogate model will update automatically.

## 4. Evolution Design Experiment with Batik Style Patterns

We collected a large number of batik shape patterns to analyze their characteristics, as shown in Figure 3. A batik shape tablecloth is shown as an example in Figure 3a. Generally, a batik shape pattern can be divided into a border pattern, which is shown in Figure 3b, and the main pattern, which is shown in Figure 3c. The main batik pattern can be separated to extract a specific pattern structure, as shown in Figure 3d.

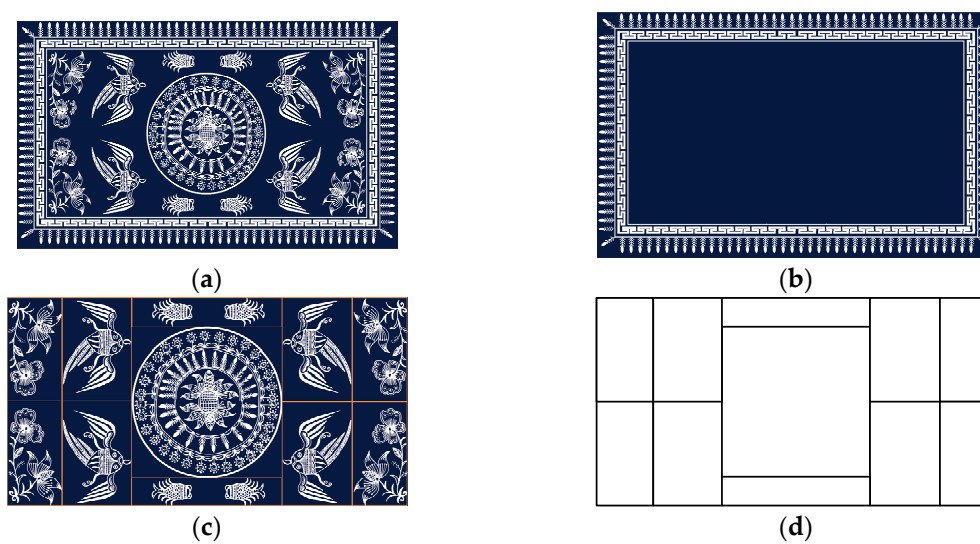

**Figure 3.** Representation of the composition of a batik shape pattern. (**a**) Batik shape tablecloth, (**b**) border pattern of a batik shape tablecloth, (**c**) main pattern of a batik shape tablecloth, and (**d**) pattern structure of a batik shape tablecloth.

### 4.1. Individual Codes

Customers' display performance requirements consist of several sub-requirements $\{d_1, d_2, \ldots, d_n\}$. In this paper, display performance is coded according to the customer's pattern structure, the specified pattern type, and the pattern size requirements. The pattern structure requirement is taken as an example, as shown in Figure 4. The L module is the border pattern; modules C, E, and I are both mainly patterns that mostly reflect the meaning of the pattern. The pattern type is specified by the customer; other modules are decorative patterns. As the batik pattern style is mostly two-square continuous or four-square continuous, eight patterns were randomly selected from the decorative pattern library, and they were rotated to a fixed angle for the decorative module gene library. The main pattern modules C and D (dragon patterns) and module E (bronze drum patterns) are taken as examples. The gene pool of each module included eight patterns, the values of coded decimals ranged from 0 to 7, which was converted to 3-bit binary code, then the search space contained $8^{\wedge}12 = 2^{36}$ candidate solutions.

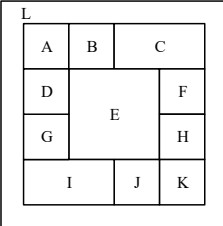

**Figure 4.** Representation of pattern's structure.

*4.2. Experimental Environment and Parameters Setting*

To measure the performance of IGA-BPFIF, IGA-T, interactive genetic algorithms with individual's interval fitness (IGA-IIF) in Shi et al. [27], and the neural network surrogate models based on individual's interval fitness in interactive genetic algorithms (IGA-NNISF) in Gong et al. [30] were compared. The parameters settings are shown in Table 1. We set the same control parameters for all four algorithms. The evolution parameters setting was completed as follows. The initial population size was $M = 9$; the crossover probability and mutation probability were set to $p_c = 0.85$ and $p_m = 0.05$ through multiple experiments, respectively; the selection operator was set as the selection of roulette with interval probability in Shi et al. [27]; the crossover operator was set as a two-point crossover; and the mutation operator was set as a single point mutation. The process for setting the surrogate model parameters was as follows. Since the pattern structure included 12 modules, the nodes of input layer $i = 12$ and the output were $F_a(x_i)$ and $F_b(x_i)$. With a normalization procedure, the nodes of the output layer were $o = 2$ and the nodes of the hidden layer were $h = 7$ according to $h = \frac{i+o}{2}$ in Lin et al. [29]. The modified fitness was determined according to the data evaluated by a human user who is trying to determine the solution to Equation (2) of the labeled samples. The labeled samples were divided into $k = 3$ parts. Each piece of datum $D(k) = \{(x_i, \widetilde{F}(x_i)), i = 1, 2, \ldots, N_k\}$ was selected in turn as the test datum and the remaining two parts as the training data to initialize the surrogate model. The MSE precision threshold was set to $e = 0.01$. The rest of the parameters in IGA-BPFIF were set as follows: $f_c(x_i)$ is composed of style $f_1$, color $f_2$, and layout $f_3$, and is standardized by domain $[f_a(x_i), f_b(x_i)] \triangleq [0, 100]$. Since the traditional batik shape patterns are all white flowers on a blue background, the color $f_2$ corresponds to weight $w_2 = 0$. If more emphasis is placed on pattern style, the weight $w_1$ of this factor increases. $f_3$ is calculated according to Equations (3)–(5). The weights of $w_{11}$, $w_{12}$, and $w_{13}$ are all $1/3$.

**Table 1.** Parameters settings.

| Evolution Parameters Setting | | | Surrogate Model Parameters Setting | | | | | Remaining Parameters in IGA-BPFIF | | | | | |
|---|---|---|---|---|---|---|---|---|---|---|---|---|---|
| $M$ | $p_c$ | $p_m$ | $i$ | $o$ | $h$ | $k$ | $e$ | $w_1$ | $w_2$ | $w_3$ | $w_{11}$ | $w_{12}$ | $w_{13}$ |
| 9 | 0.85 | 0.05 | 12 | 2 | 7 | 3 | 0.01 | 1/2 | 0 | 1/2 | 1/3 | 1/3 | 1/3 |

To set the terminal condition, the terminal evolutionary generation was set as the number of individuals that were evaluated by the user, which was 270. This means that if the users had not found their satisfied individual after 30 evolution generations without applying any surrogate model, the system automatically ends. If users find a satisfactory solution before the terminal evolutionary generation is met, they can manually end population evolution.

The system was programmed with PyQt 5, which was developed by Riverbank Computing in Wales, England, and the operating system interface is shown in Figure 5. The user clicks the initialization button, the system automatically generates the initial evolution population, and provides a display. Then, a human user evaluates the evolutionary individuals according to their preferences and clicks the next generation button, and the population conducts the selection, crossover, and mutation operations to generate new populations. If the human user's evaluation generations exceed five,

the agent model can be selected to replace the user evaluation, which means there are more than 45 labeled samples. The system takes the evaluation results of the first five generations of individuals, which means 45 labeled samples are automatically taken as the initial training data. Then, the system divides the samples into three parts to adopt K-fold cross validation to train the BP neural network cognitive surrogate model. The system estimates the adaptive value of each generation of evolution individuals, and the population carries out genetic operations to generate new populations. The surrogate model represents the optimal individual of each generation and the predicted adaptive value after nine generations. If the predicted value deviates from the user's expectations, the user can submit the evaluation again, and the system will update the training data set automatically. If $MSE > 0.01$, then the model will update automatically. The process repeats until the human user finds a satisfactory evolutionary individual.

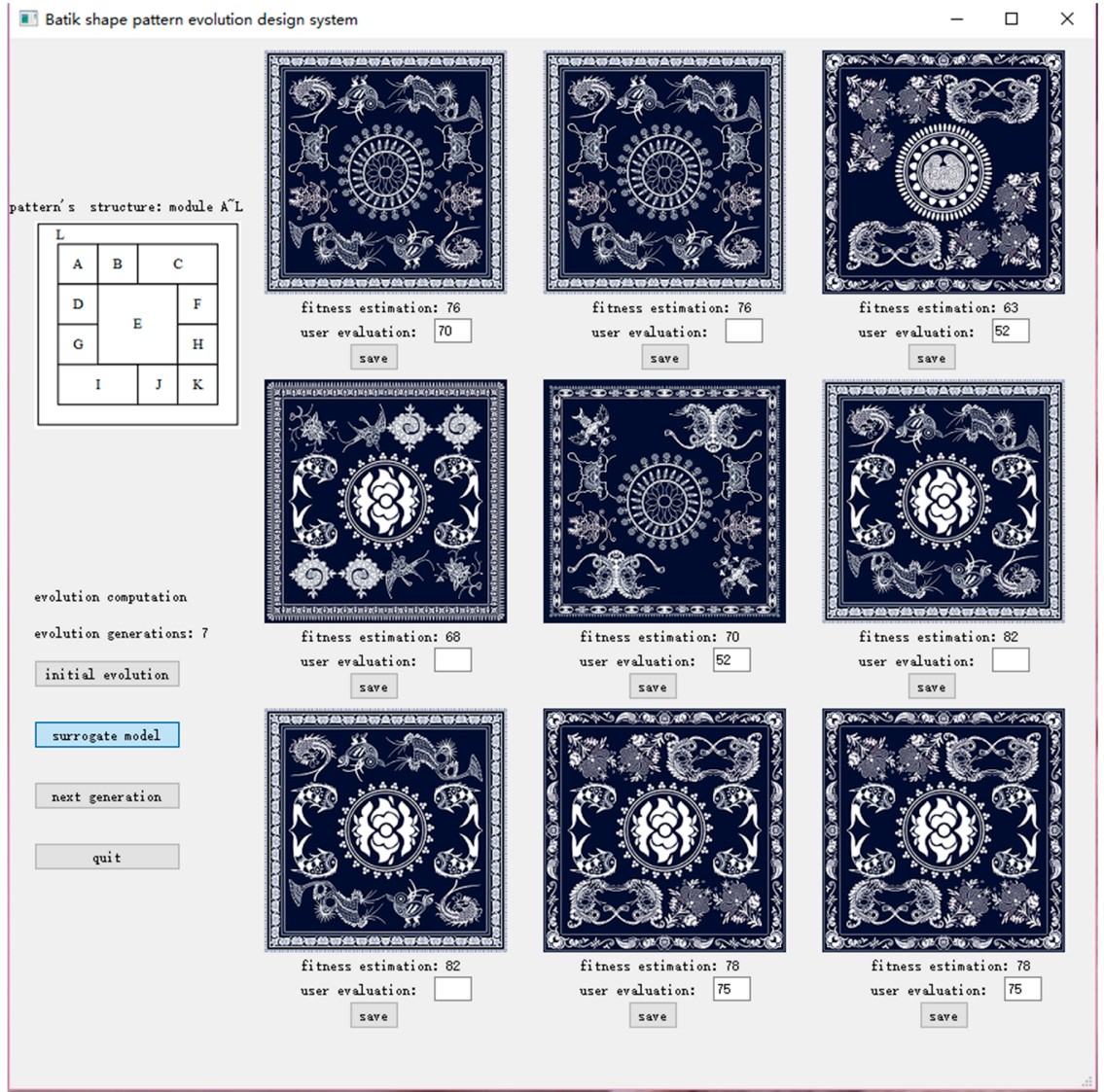

**Figure 5.** The system interface.

*4.3. Results*

For the purpose of this paper, a satisfied individual human user is the result of the experiment. Table 2 displays a subject's preferences for each algorithm in the 10th generation as the optimal result for each algorithm in the 10th generation. The optimal individual fitness for IGA-BPFIF was higher than IGA-T, IGA-IIF, and IGA-NNISF.

**Table 2.** Optimal result in the 10th generation.

| Algorithm | IGA-T | IGA-IIF | IGA-NNISF | IGA-BPFIF |
|---|---|---|---|---|
| Optimal Individual |  |  |  |  |
| Fitness | 61 | 68 | 70 | 84 |

Both the proposed algorithm, IGA-BPFIF, and IGA-NNISF selected the surrogate model after the fifth generation of evolution. When algorithms converge, the performance differences of IGA-BPFIF, IGA-T, IGA-IIF, and IGA-NNISF were compared from three aspects: the total number of iterations in each evolution, the number of individuals evaluated by the user, and the number of searched individuals in the system. As subjects, 25 graduates were randomly selected and used with the above four different algorithms to illustrate the general applicability of our method. The contrast indicators data were respectively recorded, as shown in Table 3. The average values (Avg.) and variance scores (Var.) were calculated for the three aspects above for the four different algorithms, and the comparison results are shown in Table 4. The IGA-IIF generations when the algorithm converges were fewer than IGA-T, indicating that IGA-IIF could minimize user fatigue to some extent. The number of individuals evaluated by the user of IGA-NNISF was significantly lower than IGA-IIF, indicating that IGA-NNISF could better minimize user fatigue; however, the IGA-NNISF generations were no better than IGA-IIF when the algorithm converged because IGA-NNISF applied the interval fitness in IGA-IIF, and due to the surrogate model, a user could find the satisfactory solution with more than 30 generations. The average evolution generations and the average number of individuals evaluated by users of IGA-BPFIF accounted for 35.20% and 21.74% of IGA-T, 39.71% and 24.38% of IGA-IIF, and 38.13% and 83.24% of IGA-NNISF, respectively. For a user who found a satisfactory item on IGA-BPFIF, the human–computer interaction operation of this algorithm was greatly reduced compared with the other algorithms mentioned above. Then, the number of searched items in the system increases significantly. The average number of searched individuals in the system accounted for 158.75% of IGA-T, 177.99% of IGA-IIF, and 170.90% of IGA-NNISF. To a certain extent, this reflects the effectiveness of the algorithm in improving the quality of individual evolution and reducing user aesthetic fatigue.

**Table 3.** Contrast indicators comparison.

| Algorithm | Contrast Indicators | 1 | 2 | 3 | 4 | 5 | ... | 22 | 23 | 24 | 25 |
|---|---|---|---|---|---|---|---|---|---|---|---|
| IGA-T | Evolution generations | 27 | 30 | 29 | 30 | 25 | ... | 30 | 28 | 29 | 27 |
| | Number of individuals evaluated by the user | 243 | 270 | 261 | 270 | 225 | ... | 270 | 252 | 261 | 243 |
| | Number of searched individuals | 243 | 270 | 261 | 270 | 225 | ... | 270 | 252 | 261 | 243 |
| IGA-IIF | Evolution generations | 24 | 25 | 30 | 26 | 25 | ... | 27 | 24 | 25 | 23 |
| | Number of individuals evaluated by the user | 216 | 225 | 270 | 234 | 225 | ... | 243 | 252 | 270 | 243 |
| | Number of searched individuals | 216 | 243 | 270 | 234 | 225 | ... | 243 | 252 | 270 | 243 |
| IGA-NNISF | Evolution generations | 25 | 26 | 32 | 25 | 24 | ... | 22 | 24 | 35 | 22 |
| | Number of individuals evaluated by the user | 65 | 66 | 72 | 65 | 64 | ... | 62 | 64 | 75 | 62 |
| | Number of searched individuals | 225 | 234 | 288 | 225 | 216 | ... | 198 | 216 | 315 | 198 |
| IGA-BPFIF | Evolution generations | 8 | 9 | 9 | 10 | 12 | ... | 8 | 7 | 10 | 9 |
| | Number of individuals evaluated by the user | 57 | 51 | 55 | 56 | 62 | ... | 52 | 51 | 54 | 57 |
| | Number of searched individuals | 243 | 324 | 324 | 405 | 567 | ... | 243 | 162 | 405 | 324 |

**Table 4.** Algorithm performance comparison.

| Algorithm | Evolution Generations | | Number of Individuals Evaluated by the User | | Number of Searched Individuals | |
|---|---|---|---|---|---|---|
| | Avg. | Var. | Avg. | Var. | Avg. | Var. |
| IGA-T | 28.12 | 2.36 | 253.08 | 191.16 | 253.08 | 191.16 |
| IGA-IIF | 25.08 | 5.16 | 225.72 | 417.96 | 225.72 | 417.96 |
| IGA-NNISF | 26.12 | 16.94333 | 66.12 | 16.94333 | 235.08 | 1372.41 |
| IGA-BPFIF | 9.96 | 4.04 | 55.04 | 28.12333 | 401.76 | 26,506.44 |

For the convergence performance of IGA-BPFIF, Figure 6 compares the fitness of evolutionary individuals. The y-coordinates in Figure 6a,b represent the average fitness of all evolutionary individuals $F_c(t) = \sum\limits_{i=1}^{9} F_c(x_i)$ and the average width of interval fitness, respectively. The interval fitness of the evolutionary individual is determined by the upper and lower limits of the interval $F_c(x_i) = \frac{1}{2}(F_a(x_i) + F_b(x_i))$; the x-coordinate represents the number of iterations. The four curves represent the trend of each generation of the evolution individuals of IGA-T, IGA-IIF, IGA-NNISF, and IGA-BPFIF. As can be seen from Figure 6a, the average fitness of IGA-BPFIF gradually increased with the increase in evolution generation, indicating that the evolution direction conforms to the cognitive rule. Figure 6b shows that although the average width of the interval fitness fluctuates a little, the overall trend gradually decreased, indicating that the uncertainty of user evaluation gradually decreased. As IGA-NNISF adopts the interval fitness in IGA-IIF, the average fitness was the same.

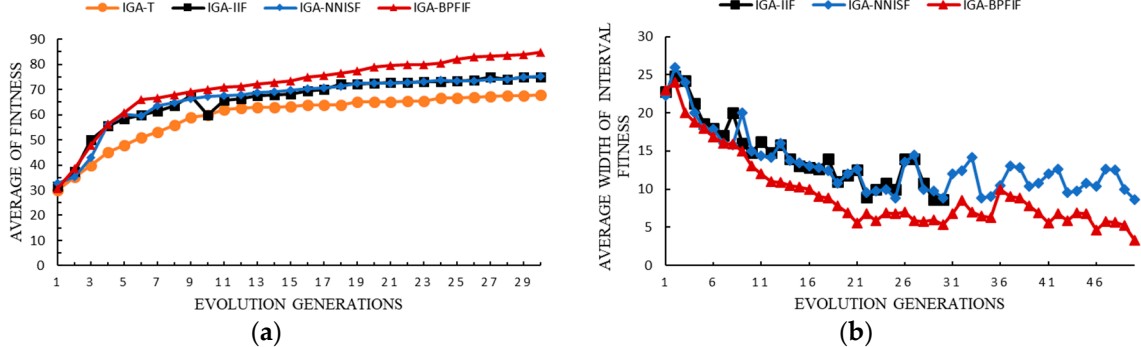

**Figure 6.** Comparison of the fitness of evolutionary individuals: (**a**) average of fitness and (**b**) average width of interval fitness.

Figure 7 compares the number of different individuals. The number of individual variations of the four algorithms increased with the increase in the number of generations. After 20 iterations, the increasing number of different individuals represented by IGA-T was no longer significant. After 25 iterations, the increasing number of different individuals represented by IGA-IIF was no longer obvious; however, IGA-BPISF experienced significant growth in the number of different individuals. This shows that IGA-BPFIF was more complex than the other three algorithms in terms of time complexity. IGA-BPFIF had more opportunities to find satisfactory solutions. By comparing the evolutionary generations under the algorithm convergence condition in Table 1, the search efficiency of this algorithm was shown to be higher than that of the other algorithms. The reason is that IGA-BPFIF modified the adaptive value of the different individual through the user's cognitive hesitation, which significantly increased the number of different individuals in each generation, thereby effectively improving the search efficiency of IGA-BPFIF. Therefore, to a certain extent, the results showed that the proposed algorithm can modify the adaptive value of the evolving individual by constructing a quantitative system for aesthetic evaluation and users' cognitive hesitation, which is conducive to reducing user evaluation noise.

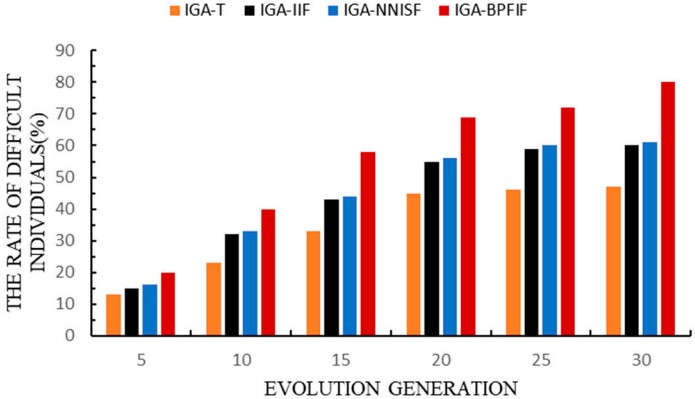

**Figure 7.** Comparison of the number of different individuals.

Individuals with fitness greater than 70 were regarded as satisfactory solutions. Figure 8 provides a satisfaction comparison of the four algorithms. When the number of generations was less than 30, the number of satisfying solutions for IGA-BPFIF was greater than those for IGA and IGA-IIF, and when the number of generations was more than 30, the number of satisfying solutions for IGA-BPFIF was 40% more than IGA-NNISF. Therefore, our algorithm effectively improved the success rate of IGA and the satisfaction degree of the users.

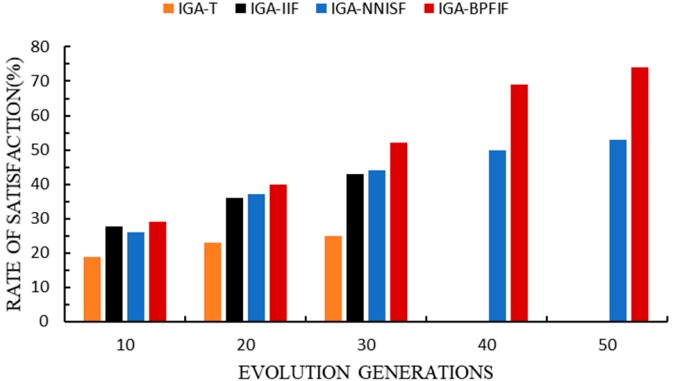

**Figure 8.** Comparison of satisfaction.

## 5. Discussion

Set $T$ as the number of evolutionary generations when the algorithm converges, and $T_1$ and $T_2$ as the number of evolution before and after the application of surrogate model, respectively; then, $T = T_1 + T_2$. In IGA-T and IGA-IIF, the number of individuals evaluated by the user and the total number of system evaluation individuals were both $T \times M$. $T$ was certain due to $T_2 \times (M - 1) > 0$. As a result, IGA-NNISF could effectively reduce the number of individuals evaluated by the user. For IGA-BPFIF, the user evaluated the number of individuals $T_1 \times M \leq N \leq T \times M$ and the number of searched individuals by the system was $T_1 \times M + T_2 \times M^2$. Since the surrogate model was used once, it was necessary to evolve the individuals $M$ times and the system presents the optimal individual in each generation. If the predicted fitness of the individuals deviated from the user expectations, user evaluation was needed. Therefore, when the number of searched individuals by the system was certain, $T_2$ in IGA-BPFIF was $\frac{1}{M}$ in IGA-NNISF. Set $T_2 = t$. Due to $N - t \times M \leq 0$, the number of individuals evaluated by a user in IGA-BPFIF was lower than in IGA-NNISF. As the total number of searched individuals in IGA-BPFIF increases exponentially, the fitness function was constructed using the quantitative rules of aesthetic evaluation and the user's cognitive hesitation to form the fuzzy interval fitness. Therefore, satisfactory individuals were more likely to be found. In summary,

compared with the above three methods, the algorithm in this paper could reduce evaluation noise, user fatigue, and effectively improve the IGA optimization performance to some extent.

## 6. Conclusions

In this paper, the fitness was measured using the aesthetic evaluation function combined with user cognitive hesitation to analyze user cognition. The surrogate model based on a BP neural network is trained using K-fold cross-validation. By measuring MSE, user preferences were identified and the surrogate model was continuously updated to guarantee performance. Experimental results showed that the algorithm could reduce user evaluation fatigue to a certain extent. Future work will focus on improving designs in terms of reusability and speed, better alignment with customer preferences, and a more effective aesthetic evaluation system optimization method will be introduced. Appropriate evolutionary strategies are still needed to accelerate the algorithm convergence.

**Author Contributions:** J.L. and W.P. administrated the project; M.Z. conceived the idea and researched the theme; M.Z. and X.L. designed and performed the experiments. J.L. and M.Z. wrote and revised the manuscript.

**Funding:** This research was supported by the Natural Science Foundation of China (Nos. 51865004, 2014BAH05F01) and the Provincial Project Foundation of Guizhou, China (Nos. [2018]1049, [2016]7467).

**Acknowledgments:** The authors would like to convey our heartfelt gratefulness to the reviewers and the editor for the valuable suggestions and important comments which greatly helped us to improve the presentation of this manuscript.

**Conflicts of Interest:** The authors declare no conflict of interest.

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
