# Peer review of "Interactive Genetic Algorithm Oriented toward the Novel Design of Traditional Patterns"

_information, doi:10.3390/info10020036_

Reviewer 1 Report

In figure 1 in the part of aesthetic fatigue there is mistake in input and output the flowchart. The ANN and GA Parameters should be illustrated in Table to better understanding. In my experience after 7 to 9 iterations the user fill tired even if just evaluate 5 individuals. in each iteration. However In fig6 the numbers of iterations is up to 46? Please explain that how ANN works in first iteration just with 5 evaluated in individuals. Explain about the number of hidden layers, test data, evaluated data and training data. It should be consider that using Historical data is not allowed in term of prefer predicting cause of it differ from on user to another. Why each method was run by 10 subjects independently for 10 times? Is it sufficient to comparing method? How the user fatigue is measured? I can’t see any statistical analysis to show the difference between the methods. In line 208 sentence” As the construction of the surrogate model is based on user’s historical evaluation information” when the prefer belongs to user personality and differ from one person to another how can user’s historical evaluation information can be useful?

Author Response

Dear Reviewer 1:

Thank you for giving us constructive suggestions which would help us both in English and in depth to improve the quality of the paper. Here we submit a new version of our manuscript with the title “information-403492-revision”, which has been modified according to the reviewers’ suggestions. Efforts were also made to correct the mistakes and improve the English of the manuscript. We mark all the changes by the "Track Changes" function in Microsoft Word in the revised manuscript.

Sincerely yours,

The authors of the manuscript

----------------------------------------------------------------------------------------------------------------------

The following is a point-to-point response to the comments.

Point 1: In figure 1 in the part of aesthetic fatigue there is mistake in input and output the flowchart.

Response 1: I sincerely apologize for my negligence of figure 1. I have already changed.

Point 2: The ANN and GA Parameters should be illustrated in Table to better understanding.

Response 2: Your suggestions have been adopted and I have illustrated the parameters in table 1 in the in the present paper. The descriptions of parameters have been adjusted to better understanding. (page 8 line 255-277)

Point 3: In my experience after 7 to 9 iterations the user fill tired even if just evaluate 5 individuals in each iteration. However in fig6 the numbers of iterations is up to 46?

Response 3: The terminal condition setting: The terminal evolutionary generation is set as the number of individuals are evaluated by the user were 270, which means if the users have not found out their satisfied individual after 30 evolution generations without applying any surrogate model, the system will automatically end. If users find out their satisfied solution before the terminal evolutionary generation is met, they could manually end population evolution.

In IGA-IIF, the max generation is 30, however, in IGA-NNISF and IGA-BPFIF, because of the surrogate model, so the max generation is more than 50.

Point 4: Please explain that how ANN works in first iteration just with 5 evaluated in individuals.

Response 4: If the human user’s evaluation generations exceed five, the agent model can be selected to replace the user evaluation, which means the labeled samples are more than 45. Then, labeled samples are divided into three parts for cross-training to train ANN.

Point 5: Explain about the number of hidden layers, test data, evaluated data and training data.

Response 5: Your suggestions have been adopted and I have adding details about the number of hidden layers, test data, evaluated data and training data. (Page 8 line 260-267)

Point 6: It should be consider that using Historical data is not allowed in term of prefer predicting cause of it differ from on user to another.

Response 6: I am very sorry about the descriptions are not rigorous enough to make you feel confused. A user who try to find out the satisfied individual needs to evaluate 9*5=45 individuals before he/she can select the surrogate model. Historical data is the evaluated data generated by the user. So I have changed the descriptions to better express the thoughts.

Point 7: Why each method was run by 10 subjects independently for 10 times? Is it sufficient to comparing method?

Response 7: Your suggestions have been adopted and the number of subjects has increased to 25 to enforce the science of the study.

Point 8: How the user fatigue is measured?

Response 8: The big question in IGA is the user fatigue due to man-machine operations frequently. Therefore, the fewer human-computer interactions (Number of individuals evaluated by the user), the less fatigue.

Point 9: I can’t see any statistical analysis to show the difference between the methods.

Response 9: Your suggestions have been adopted and I have adding details about the Contrast indicators comparison, shown as the Table 3 and more descriptions about the difference between the methods. (page 10 line 307-326)

Point 10: In line 208 sentence” As the construction of the surrogate model is based on user’s historical evaluation information” when the prefer belongs to user personality and differ from one person to another how can user’s historical evaluation information can be useful?

Response 10: I am sorry again for my trouble saying. Your suggestions have been adopted, just like point 6, a user’s historical evaluation information is just he/she own evaluated data. I have changed the descriptions to better express the thoughts. (page 6 line 192page 6 line 213)

Thank you again for my paper with my sincere appreciation! 

Reviewer 2 Report

Overview: The paper proposes a generative design methodology based on an interactive genetic algorithm and BP neural network to automatically create ethnic patterns. This is an interesting paper and the authors well address both benefits and problem of the interactive genetic algorithm. The authors suggested utilizing BP neural network to train fitness function based on user input. Experimental results are sufficient to support their algorithm. I think that the paper can be published after some mirror revisions:

[1] “No” sign below “GA operators and generate a new colony” in Figure 1 seems like a mistake.

[2] The authors showed the impact of IGA-BPFIF compared to other methods in computational costs but not the quality of the outcome. User preferences on each algorithm would expand the contribution of the paper.

Author Response

Dear Reviewer 2:

Thank you for giving us useful suggestions which would help us to improve the quality of the paper. Here we submit a new version of our manuscript with the title “information-403492-revision”, which has been modified according to the reviewers’ suggestions. Efforts were also made to correct the mistakes and improve the English of the manuscript. We mark all the changes by the "Track Changes" function in Microsoft Word in the revised manuscript.

Sincerely yours,

The authors of the manuscript

----------------------------------------------------------------------------------------------------------------------

The following is a point-to-point response to the comments.

Point 1: “No” sign below “GA operators and generate a new colony” in Figure 1 seems like a mistake.

Response 1: I sincerely apologize for my negligence of figure 1. I have already changed.

Point 2: The authors showed the impact of IGA-BPFIF compared to other methods in computational costs but not the quality of the outcome. User preferences on each algorithm would expand the contribution of the paper.

Response 2: Your suggestions have been adopted and I have displayed a user preferences on each algorithm in the tenth generation, as shown in Table 2. (page 9 line 297-301)

Thank you again for my paper with my sincere appreciation! 

Round  2

Reviewer 1 Report

I recommend to clarify why authors don't use ANFIS system and construct

their own fuzzy ANN system by mentioning that prefer prediction is very complicated 

Reviewer 2 Report

Authors sufficiently revised previous concerns.